# Evaluating the Implementation of Adolescent- and Youth-Friendly Services in the Selected Primary Healthcare Facilities in Vhembe District, Limpopo Province

**DOI:** 10.3390/ijerph21121543

**Published:** 2024-11-21

**Authors:** Mukovhe Rammela, Lufuno Makhado

**Affiliations:** Department of Public Health, Faculty of Health Sciences, University of Venda, Thohoyandou 0950, Limpopo, South Africa; lufuno.makhado@univen.ac.za

**Keywords:** evaluating, implementation, adolescent, youth, friendly, services, programme, HIV/AIDS, STI, teenage pregnancy, health care professionals

## Abstract

Background: The adolescent- and youth-friendly services (AYFS) programme has the potential to address several diverse problems within adolescents’ healthcare systems by improving the quality, accessibility, efficiency, and effectiveness of healthcare services. The country continues to suffer from structural and systemic factors that hinder the effective provision and implementation of AYFS despite its comprehensive legal and policy framework and commitment to enhancing young people’s health. Vhembe District has not been evaluated regarding the implementation of AYFS based on WHO global standards. Therefore, the objective of this study was to evaluate the implementation of AYFS against the World Health Organization (WHO) global standards for quality healthcare services for adolescents to strengthen these services in Vhembe District, Limpopo. Methods: A cross-sectional study was used to evaluate the implementation of AYFS against the WHO global standards for quality healthcare services for adolescents in Vhembe District, Limpopo. Evaluating the implementation of AYFS was conducted through questionnaires distributed to healthcare providers in the selected primary healthcare facilities in Vhembe District. For descriptive statistical analysis, research data were analysed using Statistical Package for the Social Sciences (SPSS). Results: The AYFS have been evaluated in depth across eight WHO global standards for quality health-care services for adolescents, with areas of success and areas for improvement identified. Provider competency reveals a disparity, with a majority (67.0%) of healthcare providers trained in effective communication with adolescents. In comparison, significantly fewer have received specific training in AYFS (16%) or on Pre-Exposure Prophylaxis (PrEP) (25.9%), underscoring the need for a more balanced approach to training focus. Conclusion: Research findings highlight the strengths and gaps of AYFS in Vhembe District, aligned with government and WHO priorities for adolescent health. Addressing the identified gaps is vital to ensuring that healthcare facilities are adolescent- and youth-friendly, easily accessible, and can be implemented effectively to address adolescent and youth health challenges in Vhembe District.

## 1. Introduction

In 2019, there were approximately 21 million pregnancies among girls aged 15 to 19 in low- and middle-income countries, and around 50% of these pregnancies were unintended [1]. A total of 777,000 births occurred among adolescent girls younger than 15 years [2]. According to 2019 data, 55% of unwanted pregnancies among adolescent girls aged 15 to 19 result in abortions, often performed in unsafe conditions in low- and middle-income countries [1]. In 2022 alone, 480,000 young people between the ages of 10 and 24 years were newly infected with HIV, of whom 140,000 were adolescents between the ages of 10 and 19 [3].

More than 1 million people are infected with sexually transmitted infections (STIs) each day throughout the world [4]. As a result, there are about 376 million cases of STIs every year, which include chlamydia, syphilis, trichomoniasis, and gonorrhoea, among others [5]. In addition, it is estimated that more than 500 million people are living with genital infections around the world, with herpes simplex virus (HSV) being the most common [5]. The prevalence of these STIs among young women in Vhembe District is disproportionally higher than that among young men in the same age group [6]. It is believed that this increased risk is the result of a combination of factors, including gender inequity in (sexual) relationships, early sexual debut ages, and lack of access to sexual and reproductive health (SRH) [6].

There has been a rise in teenage pregnancies, including in adolescents living with perinatal HIV, in Vhembe District, Limpopo Province [7]. More than 600 schoolgirls fall pregnant every year in Limpopo schools [8]. Limpopo Department of Health statistics indicate that there are 36 pregnant learners in each of the two secondary schools outside Thohoyandou, and 5 of these students are HIV-positive [9,10]. Another 31 learners aged between 9 and 19 from both primary and secondary schools in the same area have been infected with HIV, putting a strain on the department’s fight against the spread of HIV among the youth in Vhembe District [10]. According to the District Health Information System [DHIS] [11] 2020, Vhembe District, Limpopo Province, has reported 64,372 new HIV cases, representing approximately 11.4% of the total population. District Health Barometer (2022/23) [12] of Limpopo Province reported that Vhembe District had the highest delivery rate for 10 to 19 years in terms of the facility rate against the national average.

To curb new HIV infections and pregnancies among adolescents and teenage pregnancy, the National Department of Health (NDoH) introduced measures such as the National Adolescent & Youth Health Policy (AYHP) [13] and National Adolescent Sexual and Reproductive Health and Rights Framework Strategy (ASRH&R) [14]. The AYFS programme has been recognized as a successful model for implementing youth-friendly services within the public health system [15]. AYFS has the potential to address several diverse problems within adolescents’ healthcare systems by improving the quality, accessibility, efficiency, and effectiveness of healthcare services [16,17]. Despite the potential for AYFS to improve the quality and accessibility of adolescent healthcare services, AYFS provision and impact is relatively low in Soweto [18].

The country continues to suffer from structural and systemic factors that hinder the effective provision and implementation of AYFS despite its comprehensive legal and policy framework and commitment to enhancing young people’s health [19,20]. It has been noted that the AYFS programme is limited in Agincourt sub-district, Mpumalanga, and below the Department of Health’s target that 70% of primary healthcare facilities should provide these services [21]. Except for one clinic, participants reported that the AYFS programme was not implemented in their primary healthcare facilities [21]. Among the barriers to providing youth-oriented care reported by nurses were a lack of youth-friendly training among staff and the absence of a dedicated space for young people [21]. Among the eight facilities, four did not appear to protect the right of young people ages 12 and older to access healthcare independently [21].

Most interventions have not addressed the young and adolescent population’s specific needs due to limited resources for AYFS implementation [22,23]. Health facilities in Soweto that provide the AYFS programme do not deliver a more positive experience to young people than those that do not offer the programme [18]. Across Limpopo Province, health facilities lack Depo-Provera or Noristerat in stock, leaving women who prefer contraceptive injections frustrated [24]. As a result of the shortages, the Mopani and Vhembe districts are most severely affected, and the department has acknowledged a shortage throughout the province [24]. The literature shows little evidence of improvements in HIV, STIs, teenage pregnancy coverage, and service delivery efficiency in South Africa [25].

As part of the WHO recommendation that AYFS should be accessible, acceptable, equitable, appropriate, and effective, as well as providing quality assurance in implementing AYFS to improve adolescents’ health outcomes [16], the WHO published Global Standards for Quality Healthcare Services for adolescents to assist countries in transforming how they respond to adolescents’ health needs [16,26,27]. A range of services and commodities are offered through AYFS, including contraceptive and condom counselling, SRH education, HIV testing, and STI screening [28]. Ensuring that AYFS facilities adhere to programme guidelines, comprehensive services can be provided to a defined population of adolescents in a comfortable setting.

The implementation of the AYFS programme in Vhembe District has not been evaluated. An effective programme evaluation involves applicable, feasible, ethical, and accurate procedures for improving and accounting for public health interventions. As a result, this study aimed to evaluate the implementation of the AYFS programme in Vhembe District against the WHO global standards to strengthen these services [29].

## 2. Materials and Methods

### 2.1. Study Design

A cross-sectional study was conducted to evaluate the implementation of AYFS against the WHO global standards for quality healthcare services for adolescents to strengthen these services in Vhembe District, Limpopo. Evaluating the implementation of AYFS was conducted through questionnaires distributed to healthcare providers in the selected primary healthcare facilities in Vhembe District.

### 2.2. Study Setting

The study was conducted in Vhembe District, Limpopo Province, in South Africa. According to the District Health Information System [DHIS] annual report of 2020 [11], Vhembe District, Limpopo Province, has reported 64,372 new HIV cases, representing approximately 11.4% of the total population. The Limpopo Province District Health Barometer (2022/23) [12] reported that Vhembe District had the highest delivery rate of 10- to 19-year-olds in terms of the facility rate against the national average.

### 2.3. Study Population

The study population included clinical staff (nurses) and non-clinical support staff (community healthcare workers, counsellors, and mentors) within the selected healthcare facilities in Vhembe District, Limpopo Province.

### 2.4. Study Sampling

Simple random sampling was used to sample 112 healthcare providers (nurses, counsellors, youth champions, mentors, and community healthcare workers) working in the sexual and reproductive program from the selected healthcare clinics that provide adolescent- and youth-friendly services in governmental clinics. This study used a Raosoft online calculator to calculate sample sizes for 112 out of 156 healthcare providers (with a 5% margin of error, 95% confidence level, and 50% distribution response).

### 2.5. Data Collection Tools

A pre-tested, structured, self-administered questionnaire was used to collect data from respondents. The pre-test was conducted on 5% (*n* = 45) of the study sample. The results and the facility of the pre-test were not included in the main study. The results of the pilot study were considered reliable. This tool has been adapted from the WHO global standards for quality healthcare services for adolescents, scoring sheets for data collection [26,27], and is intended to clarify and simplify the AYFS evaluation process. The WHO global standards scoring guideline for data analysis is commonly used in developing countries and was used to assess adolescent-friendly health clinics in two of India’s largest states [30,31]. The questionnaire (developed in English and then translated to Venda and Tsonga) had two sections, demographics and 08 AYFS standards, and 21 criteria. Each question was measured by 1 = Yes or 0 = No and took 20 to 30 min to administer. A response rate of 100% was achieved. The researcher collected the questionnaires at the end of the survey.

### 2.6. Data Management and Analysis

For descriptive statistical analysis, research data were analysed using Statistical Package for the Social Sciences (SPSS) Version 29.0 (IBM Corp, Armonk, NY, USA). Frequencies and percentages were used as descriptive statistics for the demographic characteristics and healthcare providers’ implementation scores based on the WHO global standards for quality healthcare services for adolescents: scoring sheets for data collection.

## 3. Results

### Profile of Health Service Providers

Most participating health service providers were female, 89.3%, and 10.7% were males (see Table 1). Participants’ occupations include 26.8% community health workers, 8.0% HTS counsellors, 55.4% nurses, 4.5% social workers, and 5.3% mentors. Most healthcare service providers (49.1%) said they had been working with adolescents or young people for over 11 years, and 24% said they had been working with them for between 6 and 10 years, signifying many years of experience working with adolescents and young people.

Different sub-districts under Vhembe District were represented, including Musina 38.4%, Makhado 19.6%, Thulamela 24.1%, and Collins Chabane 17.9%. Overall, the results showed that most of the population resided in Musina.

The study findings provide an in-depth assessment of the current state of AYFS implementation across WHO global standards (see Table 2) and outline areas of success and areas needing improvement.

Standard 1: Adolescent Health Literacy—In terms of adolescent health literacy, the findings indicate a moderate engagement with the AYFS program, with approximately one-third of participants (30%) utilizing informational, educational, and communication (IEC) materials in their waiting area. In addition to participating in community outreach activities, 36% of respondents demonstrated a reasonable level of interest and interaction with the resources provided. As evidenced by the overall percentage score below 31%, healthcare providers also demonstrate ineffective implementation of AYFS signage and branding.

Standard 2: Community Support—There was relatively low community engagement, with only 21.5% of healthcare providers engaging with parents or guardians of adolescent patients and 16% using supportive materials to promote AYFS values. Healthcare providers’ participation in school-based engagement is slightly higher at 43.8%, indicating the need to strengthen community ties and support systems.

Standard 3: Appropriate Package of Services—The awareness of the AYFS service package among healthcare providers is approximately 48.2%, indicating an opportunity for improvement in disseminating and training the essential services to ensure effective implementation and uptake of the AYFS programme.

Standard 4: Providers’ Competencies—Healthcare providers’ competency reveals a disparity, with the majority (67.0%) trained in effective communication with adolescents. In contrast, significantly fewer participants have been trained on AYFS (16%) and Pre-Exposure Prophylaxis (PrEP) (25.9%), indicating the need for a more balanced approach to training focus.

Standard 5: Facility Characteristics—Although facilities demonstrate 100% commitment to accessibility by operating on weekends, there are insufficient specific time slots for adolescents (46.4%), limiting their access to services.

Standard 6: Equity and Non-Discrimination—Most participants effectively implemented the required equity and non-discrimination standards. It was found that 80% of the privacy guidelines protecting the confidentiality of adolescents were adhered to. Healthcare providers are committed to equity and non-discrimination by following guidelines for ensuring equitable services for all adolescents (71.4% and 83.9%, respectively). However, over 26.8% of healthcare providers reported denying adolescents services due to age-related consent issues, indicating a significant challenge in service delivery that warrants immediate attention. Adolescent- and youth-friendly services are based on a strong foundation, which suggests that efforts should be focused on enhancing these services’ reach, effectiveness, and equity.

Standard 7: Data and Quality Improvement—It is worth noting that while more than half of healthcare providers (51.8%) receive training in monitoring AYFS performance and 56.2% participate in data review sessions, only 32.1% meet the national benchmarks set by the Department of Health. It demonstrated a need for improvement, as their percentage scores ranged from 40% to 60%.

Standard 8: Adolescent Participation—The involvement of adolescents and vulnerable groups in planning, monitoring, and evaluating health services is limited, with participation rates of 25.9% and 34.8%, respectively. These results underscore the need for more inclusive strategies to enhance engagement and ensure that services meet the needs of all young individuals. These findings paint a clear picture of the current state of AYFS implementation, highlighting the necessity to address gaps in provider training, community involvement, and adolescent participation to enhance service delivery.

## 4. Discussion

The adolescent- and youth-friendly services (AYFS) programme has not been evaluated in Vhembe District based on the WHO global standards for quality healthcare for adolescents. While studies have been conducted in Limpopo Province on challenges faced by nurses implementing AYFS [32], the programme effectiveness has been assessed internationally [30] and in other South African provinces, such as Gauteng and Northwest [33]. This study evaluates the AYFS implementation in Vhembe District, identifying areas of success and improvement according to WHO standards. The findings reveal effective implementation in some areas, with others needing attention. For instance, similar to findings in Gauteng, no facility in Vhembe met the minimum AYFS recognition criteria in all sub-districts [33].

The findings underscore strengths and critical areas for improvement in implementing AYFS in Vhembe District. While specific standards, such as facility characteristics and equity in service delivery, demonstrate encouraging progress, the program’s overall effectiveness is hindered by persistent gaps in crucial areas like adolescent health literacy, community engagement, and healthcare provider training. These gaps reveal systemic issues that limit the program’s ability to fully meet the WHO’s global standards for quality healthcare for adolescents. Limited awareness and training among providers, insufficient adolescent participation, and weak community support highlight a lack of integrated approaches essential for sustainable service delivery. Moreover, the inconsistencies in monitoring and evaluation mechanisms, reflected in unmet national benchmarks, indicate a need for capacity building and better data-driven decision-making. Strengthening these areas through targeted interventions, such as comprehensive training programs, improved communication strategies, and greater involvement of adolescents and communities, can enhance the reach, quality, and equity of AYFS. Addressing these challenges holistically will ensure that healthcare services are more responsive to the needs of adolescents, thereby fostering better health outcomes [34,35,36].

Equity and non-discrimination are core components of adolescent health services. Staff training is essential to uphold these principles, enabling equitable access to services irrespective of socio-economic or cultural backgrounds. Adherence to informed consent guidelines and standard operating procedures (SOPs) significantly correlated with service quality in Vhembe. Healthcare providers demonstrated high compliance with data protection and confidentiality policies. However, challenges safeguarding adolescent rights, such as privacy and non-discrimination, were noted in other settings, including India [30,37]. Similarly, a study in Kenya revealed that 53% of service providers were unaware of youth-friendly policies, with 75% of facilities lacking relevant implementation guidelines [38]. Training and education have effectively addressed these gaps and improved adherence to SOPs [39,40].

Healthcare providers actively attended facility data interrogation sessions, underscoring their importance in improving patient outcomes. Nonetheless, certain aspects of the AYFS programme, such as information, education, and communication (IEC) materials, facility signage, and branding, need enhancement. The absence of IEC materials in most service outlets limits adolescents’ health literacy and awareness among families and communities about their health needs [41]. Evidence indicates that IEC materials can reduce self-stigma and increase community acceptance [42]. The lack of adolescent participation in planning, monitoring, and evaluation processes, a challenge observed in Vhembe and Kenya [38], further underscores the need for meaningful youth involvement in decision-making to improve service delivery [43].

This study highlighted gaps in healthcare providers’ familiarity with the AYFS package compared to their counterparts in East Africa [38], leading to low implementation rates in Vhembe District. Addressing these gaps requires regular training for healthcare providers to build their capacity to deliver AYFS effectively [44,45]. Similar challenges were noted in Gauteng, where disconnects between programme policies and facility-level performance were attributed to poor stakeholder communication and a lack of understanding of policy objectives [33]. Aligning AYFS guidelines with standards and service packages is necessary for improved implementation.

Limited training on AYFS and related areas, such as HIV oral Pre-Exposure Prophylaxis (PrEP), further hinders programme success. Insufficient training leads to confusion, mistakes, and non-compliance, affecting staff morale. Similar gaps in healthcare providers’ knowledge and technical skills were reported in India [31], contrasting with Ethiopia’s adequately staffed adolescent-friendly health clinics [46]. Training programs should emphasize the impact of stereotyping and judgment on service delivery and frame PrEP as an empowering tool for adolescent girls and young women (AGYW) [47,48].

Effectively monitoring AYFS indicators is crucial for identifying service gaps and making data-driven improvements. However, inadequate training in data collection and analysis among healthcare providers in Vhembe undermines the reliability of performance indicators. Comprehensive training programs, regular workshops, and mentorship initiatives are recommended to build provider competence in data management. Improved monitoring of AYFS indicators will enable healthcare providers to address critical service gaps and enhance health outcomes for adolescents and young people.

The AYFS programme in Vhembe District can better align with WHO standards, ensuring accessible, equitable, and effective adolescent health services by addressing these challenges. Collaboration with youth and the community, as well as ongoing training and infrastructure improvements, will strengthen the programme and enhance its impact.

### 4.1. Limitations

The study was only conducted in eight (8) clinics in Vhembe District. Therefore, the results cannot be generalised to the rest of Limpopo Province. A large-scale study using standardized methodologies is required to evaluate the implementation effectiveness of AYFS clinics found in Limpopo province.

### 4.2. Implications for Effective AYFS Implementation and Future Research

Adolescent and youth-friendly services (AYFS) must be implemented effectively to ensure that its benefits are realized. This includes adequate funding, proper planning, and adequate support from stakeholders. Additionally, AYFS must be regularly evaluated and updated to remain effective. AYFS should also be assessed to ensure any unintended consequences are addressed quickly. Finally, future research on AYFS implementation should explore the effectiveness of various strategies for successful implementation.

## 5. Conclusions

Addressing the existing gaps in the implementation of AYFS is essential for the enhancement of healthcare experiences among adolescents in the Vhembe District. Strengthening AYFS necessitates a collaborative approach involving all stakeholders, thereby underscoring the importance of community engagement and effective partnerships. Improving communication between healthcare providers and parents is crucial for fostering greater awareness and involvement in adolescent healthcare. Furthermore, prioritizing training for healthcare staff will equip them with the necessary skills and knowledge to deliver high-quality services tailored to the needs of this population. It is equally important to allocate sufficient resources to support AYFS initiatives throughout all communities. Through the adoption of these comprehensive measures, the healthcare system can create a responsive environment that adequately meets the unique needs of adolescents. Ultimately, this approach is expected to lead to improved health outcomes and contribute to a healthier future for the community.

## Figures and Tables

**Table 1 ijerph-21-01543-t001:** Socio-demographic characteristics of the study participants.

Variables	Frequencies (*n* = 112)	Percentages (%)
Gender		
Men	12	10.7
Women	100	89.3
Occupation		
Community Healthcare	30	26.8
HTS Counsellor	9	8.0
Nurse	62	55.4
Social Worker	5	4.5
Mentor	6	5.3
Work experience in years		
01–05	11	9.8
06–10	27	24.1
11–20	55	49.1
21–30	14	12.5
31 and above	5	4.5
Sub-districts		
Musina	43	38.4
Makhado	22	19.6
Thulamela	27	24.1
Collins Chabane	20	17.9

**Table 2 ijerph-21-01543-t002:** Healthcare providers’ implementation scores based on the WHO global standards for quality healthcare services for adolescents: scoring sheets for data collection.

Variables	YES *n* = 112 (%)	NO *n* = 112 (%)
Standard 1: Adolescent Health Literacy		
Usage of AYFS signage and branding	34 (30.4)	78 (69.6)
Usage of AYFS IEC materials	39 (34.8)	73 (65.2)
Community outreach activities conducted	40 (35.7)	72 (64.3)
Standard 2: Community Support		
Engagement with parents, guardians, or family members of their young clients	24 (21.4)	88 (78.6)
Usage of support materials to communicate the values of implementing AYFS	18 (16.1)	94 (83.9)
Participation in school meetings to engage with learners	49 (43.8)	63 (56.2)
Standard 3: Appropriate Package of Services		
Healthcare provider’s awareness rate of AYFS package	54 (48.2)	58 (51.8)
Standard 4: Providers’ Competencies		
Health providers received training on AYFS	18 (16.0)	94 (84.0)
Health providers received training on PrEP	29 (25.9)	83 (74.1)
Health providers received training on effective communication with adolescents	75 (67.0)	37 (33.0)
Standard 5: Facility Characteristics		
Utilisation of guidelines on protecting the privacy and confidentiality of adolescents	41 (36.6)	71 (63.4)
Facilitates working over weekends	112 (100)	0 (0)
Utilisation of slot for adolescents and youth in the facility	52 (46.4)	60 (53.6)
Standard 6: Equity and Non-Discrimination Utilising guidelines/SOPs on informed consent Utilisation of guidelines/SOPs equitable services to all adolescents irrespective of their age, sex, marital status Adolescents denied services in the last 12 months due to their age, need for parental, spousal or partner consent	80 (71.4) 94 (83.9) 30 (26.8)	32 (28.6) 18 (16.1) 82 (73.2)
Standard 7: Data and Quality Improvement Trained health care providers on data collection and analysis for monitoring AYFS indicators performance Trained healthcare providers on quality improvement methodology Healthcare providers meeting the NDoH set targets on AYFS indicators Participation of healthcare providers in facility data interrogation sessions	58 (51.8) 20 (17.9) 36 (32.1) 63 (56.3)	54 (48.2) 92 (82.1) 76 (67.9) 49 (43.8)
Standard 8: Adolescent Participation Involvement of adolescents in the planning, monitoring, and evaluation of their health services Involvement of vulnerable groups in the planning, monitoring, and evaluation of their health services	29 (25.9) 39 (34.8)	83 (74.1) 73 (65.2)

## Data Availability

Data presented in this study can be obtained from the corresponding author upon request. Due to the sensitivity of the study and the ethical clearance conditions, the data are not publicly available.

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
