# Peer review of "Evaluating the Implementation of Adolescent- and Youth-Friendly Services in the Selected Primary Healthcare Facilities in Vhembe District, Limpopo Province"

_ijerph, 2024, doi:10.3390/ijerph21121543_

Round 1

Reviewer 1 Report

Comments and Suggestions for Authors

The authors present the results of the evaluation process of the Adolescent and Youth-friendly services implemented in the Vhembe District, in South Africa. As the authors acknowledged in the Limitation section of the manuscript, being this evaluation conducted only in one District in South Africa the results cannot be generalized, however they are informative for the development of local policies,

The manuscript is well written, some slight improvements in the English language are recommended.

Some minor comments.
1. In the Introduction section, please spell out the meaning of AYFS the first time they are mentioned (line 40). 

2. In the Methods section, provide  short description of the eight standards of the AYFS provided in the LoveLife AYFS Toolkit (Line 101). A reference to the Toolkit should also provided. 

3. Provide a brief explanation on how the pilot study was conducted (duration, number of staff interviewed, methos used to assess the validity of the questionnaire) (Line 104). 

4. Did the authors find any differences across sub-districts in the implementation of AYFS? It could be useful to have some additional information on possible differences 

Author Response

Manuscript: ijerph-3267437

Response to Reviewers

Dear   Gorana Belegisanin

Thank you for giving us the opportunity to submit a revised draft of the manuscript “Evaluating the Implementation Effectiveness of an Adolescent and Youth-friendly Services Programme to Address HIV/AIDS, STI and Teenage pregnancy in Vhembe District” for publication in the International Journal of  Environmental Research and Public Health.

 We appreciate the time and effort that you and the reviewers dedicated to providing feedback on our manuscript and are grateful for the insightful comments on and valuable improvements to our paper.

We have incorporated most of the suggestions made by the reviewers.  Please see below, in red, for a point-by-point response to the reviewers’

comments and concerns.  

Reviewers' Comments to the Authors:

 Comments from Reviewer 1

The authors present the results of the evaluation process of the Adolescent and Youth-friendly services implemented in the Vhembe District, in South Africa. As the authors acknowledged in the Limitation section of the manuscript, being this evaluation conducted only in one District in South Africa the results cannot be generalized, however they are informative for the development of local policies,

The manuscript is well written, some slight improvements in the English language are recommended.

Some minor comments.

Comment 1. In the Introduction section, please spell out the meaning of AYFS the first time they are mentioned (line 40).

Response 1. Thank you for pointing this out. The comment is well taken. Therefore, we have written AYFS in full the first time. Page 2, Line 52.

Comment 2. In the Methods section, provide  short description of the eight standards of the AYFS provided in the LoveLife AYFS Toolkit (Line 101). A reference to the Toolkit should also provided.

Response 2. As suggested by the reviewer, we have provide a short description of the eight WHO global standards for quality healthcare services for adolescents  and referenced it accordingly. The researcher  adopted the WHO global standards for quality healthcare services for adolescents: scoring sheets for data collection to measure AYFS implementation as the LoveLife AYFS Toolkit was developed  from WHO global standards for quality healthcare services for adolescents: scoring sheets for data collection. Page 2, Line 73-77 and page 3, line 127-133.

Comment 3. Provide a brief explanation on how the pilot study was conducted (duration, number of staff interviewed, methos used to assess the validity of the questionnaire) (Line 104).

Response 3. The method used to assess the validity was a construct validity as evidence of previous studies adopted the WHO global standards for quality healthcare services for adolescents: scoring sheets for data collection to measure AYFS implementation is provided and cited. The pre-test was conducted on 5% (n=45) of the study sample, the results and the facility of the pre-test were not included in the main study. Page 3, line 126-132.

Comment 4. Did the authors find any differences across sub-districts in the implementation of AYFS? It could be useful to have some additional information on possible differences

Response 4. AYFS was implemented uniformly across the sub-districts as they shared the best practices regarding operating on weekends and adhering to equity SOPs and guidelines. As a result, there is a shortage of trained staff in all sub-districts regarding AYFS and PrEP, and those who are trained are scattered throughout the Sub-districts. Therefore, there is no differences across sub-districts in the implementation of AYFS. Page 4-5.

Reviewer 2 Report

Comments and Suggestions for Authors

The report is attached. 

Author Response

Abstract.

Comment 1. The objective must be formulated more specifically. E.g. “The objective of this study is to evaluate the effectiveness of the AYFS programme implemented to address HIV and AIDS, STIs and teenage pregnancy in the Vhembe district in Limpopo province, South Africa”- the mention of health care professionals within the objective is not clear. - using the key headings that make up a good abstract (background, methods, results, conclusion) can be beneficial for clarity and easy identification of sections within the abstract. Currently, the abstract reads as though it has no conclusion – ends with results.

Response 1. Agree. We have, accordingly, revised the objective of this study and now it reads are follow: The objective of this study was to evaluate the implementation effectiveness of the Adolescent and Youth-friendly services Programme against the WHO global standards for quality health-care services for adolescents in Vhembe district, Limpopo. Conclusion is added and well summarized to the abstract. Page 1, line 18-24 and 30-33.

Introduction section:

Comment 2. This section of the manuscript disappoints a little. The authors have not expanded, nor mobilised sufficient literature to position the gap that this study has identified and therefore seeks to fill. The first paragraph provides global stats, which is good way to position the study by showing the challenges facing adolescents as a particular vulnerable population. However, there are no stats specific to the SA context and Vhembe as a district. This should be part of the introduction. Then transition to the adoption of AYFS by the NDoH is premature without the specific stats of SA. Reference the 1st sentence in paragraph two! What does sub-optimal mean?

Response 2. Agree. We have, revised the introduction and added   sufficient literature to position the gap that this study. The comment made regarding stats of SA is well taken. As a result, we have added stats on HIV and teenage pregnancy in the context of South Africa and the Vhembe district (According to the District Health Information System [DHIS] 2020, the Vhembe district of Limpopo province has reported 64,372 new HIV cases, representing approximately 11.4% of the total population. District Health Barometer (2022/23) of Limpopo Province reported that Vhembe District had the highest delivery of 10 to 19 years in facility rate against the national average.) on page 2, between line 47 and 50. ( Most of the page 1 has been changed.)

Comment 3. It not clear whether the introduction includes the review of literature. The authors must re-arrange the ‘introduction’ section to show the building of their argument by reviewing relevant literature. This section must draw from key authors who can provide excellent context of AYFS in SA. See authors like (not limited to):

  1. Geary, R.S. et al. (2014) ‘Barriers to and facilitators of the provision of a youthfriendly health services programme in rural South Africa’, BMC Health Services

Research, 14(1), p. 259. Available at: https://doi.org/10.1186/1472-6963-14-259.

_ Look at all of Geary’s publications (2017 etc).

Note for the second sentence in paragraph 2: There could be value in mentioning the South African strategies/policies informing HIV prevention and adolescent health:

- the National Adolescent and Youth Health Policy (AYHP)

- National Adolescent & Youth Health Policy (AYHP (2016-2020)

- the National Adolescent Sexual and Reproductive Health and Rights (ASRH&R)

Framework Strategy (2014-2019)

These policies form part of the instruments adopted at national level in SA for adolescents. The authors don’t have to write at length about these. But a few sentences to acknowledge and reference the work that has been done in SA, which has been foundational to the adoption of the AYFS programme. E.G. The AYHP policy document acknowledges that health promotion depends on providing functional and youth-friendly healthcare services. To merely say “several programs and policies” is a missed opportunity to build a stronger argument.

Response 3. As suggested by the reviewer, we have rearranged the introduction section and build argument that AYFS has the potential to address several diverse problems within adolescents' healthcare systems by improving the quality, accessibility, efficiency, and effectiveness of healthcare services while also reducing the costs [6,8]. Despite the potential for AYFS to improve the quality and accessibility of adolescent healthcare services, their success rate has been inconsistent [9,10] page 2 lin 45-72.

Comment 4. The entire second paragraph makes bold statements with just on reference. Then the closing sentence of this paragraph claims that “literature shows…” yet very little literature has been interrogated in this manuscript. Both SA literature and AYFS in other African countries is barely presented here.

Response 4. Agree. We have, revised the introduction and added sufficient literature to position the gap that this study. Page 2 line 45-78.

Comment 5. The 3rd paragraph, authors could maybe footnote the WHO framework constructs: accessible, acceptable etc. What do these mean for someone who does not know? And for the context of the study? The point is not catalogue information, but rather to use it to help you make your argument. The closing sentence in this paragraph is extremely long, and meaning gets lost. Avoid long sentences, yours are loaded! The 4th paragraph also opens with a very long sentence. Long sentences usually are full of sweeping statements -sentences that are just general and have no references or show value in a point you are trying to make. You just mention “big and important” information in passing. The authors must read wider and reference recent studies as they build context and argument in this section. I would suggest that this section be structured a bit differently:Introduction and context – paint the scene for the need for the study, explaining the stats and outline the main aims and research questions. Literature review – here you want to deep dive into literature which draws on any links between AYFS and HIV and AIDS, STI’s and Teenage pregnancy among adolescents. Other studies that have evaluated effectiveness etc. Is the focus on both male and female adolescents? HIV and STI’s can be common to both, which warrants my question. Clarify this and then say for e.g. that the focus is on females.

Do not present it as a review of literature but rather, present it as two broad arguments:

Argument 1 – showing that AYFS has the potential to address the HIV,STI’s and teen pregnancy.

Argument 2 – a counter argument to the above – so are there any other dialogues which speak to AYFS? Maybe barriers and facilitators to AYFS for example. This can form a smaller part of this section, maybe 1 or 2 clean paragraphs.

Last paragraph – your critical review of the two arguments and the position that is being put forward by the paper

This is just an example of how an argument ought to flow and what you include is directed by your main aim and research question (you don’t have to write the headings in bold in the manuscript, but it’s for you to get an idea of how this section should look/sound). You reference James et al., 2018 once. What this current study seeks to do is very similar to the James article in my opinion. The authors can take notes from that paper to re-work this section – making a clearer and a more compelling argument.

Looking at the reference list, you only use literature in your discussion but did not review to this to build a compelling argument for the reader. You convince the reader at the beginning and then use the same literature in your discussion, with some new references where needed.

From reading this, I am not convinced that you presented relevant data in this manuscript. If the aim and research question is not clear, then how would you be sure on what data to present and analyse?

Response 5. As suggested by the reviewer, we have rearranged the introduction section and build argument that AYFS has the potential to address several diverse problems within adolescents' healthcare systems by improving the quality, accessibility, efficiency, and effectiveness of healthcare services while also reducing the costs [6,8]. Despite the potential for AYFS to improve the quality and accessibility of adolescent healthcare services, their success rate has been inconsistent [9,10]. page 2 line 55-70.

Materials and methods:

Comment 5. Your methods need to fall from the aim and research question. The better articulated the question is, the more readily they can go much further in directing the nitty gritty of methods. This study evaluates the implementation of AYFS in primary healthcare facilities against defined standards to inform initiatives for strengthening these services in Vhembe. What are defined standards? Which ones? There a few when it comes to AYFS. Be specific and then say why these particular standards. I elaborate:

  1. What does this aim really intend to do? What are these defined standards? Do

you mean the 5 WHO constructs (accessibility, acceptability etc… or the 10

standards of services and management)?

  1. To inform initiatives? What initiatives? What does this mean?

This is an attempt to ‘break down’ the aim so that it can be re-worked. Take note of

the James et al., 2018 paper (and may other papers) to guide how to write your aim

etc.

Can you reference the claim that Vhembe had the highest HIV stats etc. The IDP is mentioned, but other references can valuable, especially because there is a comparison to other regions. If there is work done in this district, it would add great value for why you did this study there.

New HIV cases in the Vhembe district, Limpopo province are

reported to be at 64 372 – approximately 11.4% of the total

population (District Health Information System [DHIS] 2020:13).

This is lower than the HIV incidence in Capricorn district (70 710,

approximately 19.4% of the total population) (DHIS 2020:08), where

the high incidence rate may be because of the urban nature of the

district, which is home to fewer traditional initiation schools than

Vhembe.

This is just an example, it is not related to AYFS, but can help you to build a stronger argument even in the introduction and literature review section above. There are many more, just spend more time reading wider ☺.

Response 5. The aim of this study was to evaluate the implementation of the Adolescent and Youth-friendly services programme against the WHO global standards for quality healthcare services for adolescents in Vhembe district, Limpopo. An evaluation of Adolescent and Youth-friendly service implementation was conducted through questionnaires distributed to healthcare providers and non-clinical staff. Page 3, line 101-105.

Comment 6.  Under sampling, “112 sample sizes were sampled” – this may need editing.

Response 6. Thank you for pointing this out, edition is done and now it reads as follow: . Using the Raosoft online calculator, 112 sample sizes were calculated from the total population of 156 healthcare providers (with a 5% margin of error, 95% confidence level, and 50% distribution response). page 3, line 120-122.

-Comment 7. Under data collection tools: “each question was measured by Yes and No? Does this  mean the entire questionnaire had “yes and no” statements for the respondents to  respond to? If the questionnaire adopted the LoveLife AYFS Toolkit, then what does  yes and no mean?

- Are you also reporting results from a pilot study? This is mentioned without clarity.

- What are the tests of reliability and validity?

- What are the methods of analysis?

Response 7. As suggested by the reviewer, we have provided a short description of the eight WHO global standards of the AYFS and reference it accordingly. The researcher  adopted the WHO global standards for quality healthcare services for adolescents: scoring sheets for data collection to measure AYFS implementation. The WHO scoring sheet is designed with a yes and no to measure the implementation of AYFS as the LoveLife AYFS Toolkit was developed from WHO scoring sheet for data collection. The method used to assess the validity was a construct validity and evidence of previous studies adopted this instrument is provided and cited. The pre-test was conducted on 5% (n=45) of the study sample, the results and the facility of the pre-test were not included in the main study. page 3, line 125-132.

Comment 8. Table 1: Perhaps add an additional column and add the percentages separately from the frequency.

Table 2: The authors only talk about this table in the discussion, yet it was presented much earlier in the results section. It needs to be interpreted with more clarity. The table itself needs to be expanded with additional rows and columns to make it easier to read. The reader should not be looking for meaning in anything presented. There must be clarity and flow provided by the authors on why its presented, what does it offer and tell the reader etc.

So, is Table 1 & 2 the only represented data on this evaluation?

Response 8. Table 1 column added. Page 4 line 56.

We agree with this suggestion and have incorporated your suggestion throughout the manuscript. 

Comment 9. Discussion: This section is very long. You have not spoken about these standards at all and you mention about 2 or more in the discussion section. But what are these standards( all of them). Is it not worth mentioning then earlier and positioning your study with them in the earlier sections of this draft?

The discussion often reads as an attempt of a literature review.

A strong discussion states WHY and HOW other studies are similar or different from our study finding. Whereas, a weak one states WHAT studies are similar or different from our study finding.

Tips as you write: Is this finding unexpected? Why do you think that is? Explain it and don’t sweep it under the rug (as you do in the current draft).

Are there concordant findings with other studies? Why is it similar? How does your study finding add more to it? Are there discordant findings with other studies? Or mixed? Why is it different? Different population maybe? Sample size too small maybe?

My study is better because…

Suggested Discussion structure:

Paragraph1: The answer to your research question (key results)

Middle paragraph: Discuss findings in a sequence

Final paragraph: Limitation, Strengths, application/future direction

Response 9. As suggested by the reviewer, we have reworked the discussion section and breakdown the WHO global standards for quality healthcare services of adolescents as we discuss the findings of the study. However, the first part of the discussion covers the best practices and areas for improvement then follows.  Table 1 column added. Page 4 line 56.

We agree with this suggestion and have incorporated your suggestion throughout the manuscript. Concordant findings with other studies summarized and cited accordingly the entire discussion. Page 6 line213-325.

Round 2

Reviewer 2 Report

Comments and Suggestions for Authors

The paper has 'new' information plugged into it in red. However, this paper is difficult to read and understand the authors' objective. The new information seems to have been pasted without re-reading if it flows. This makes the manuscript difficult to read. Some sentences are not referenced, and some sound incomplete. The entire manuscript could do with heavy editing. The previous report seems to have been rushed through without addressing the review carefully. From the abstract, it is challenging to clearly understand what this paper wants to do. It is framed as if the authors are intending to write a thesis. I attached a document with a few notes.  

Author Response

Evaluating the Implementation Effectiveness of an Adolescent and Youth-friendly Services Programme to Address HIV/AIDS, 3 STI and Teenage Pregnancy in Vhembe District.  The title has changed, but is still not clear. “IMPLEMENTATION EFFECTIVENESS?The topic has been revised as follows:  Evaluating the Implementation of Adolescent and Youth-friendly Services in the selected primary healthcare facilities in Vhembe District, Limpopo province.

 Abstract.  

Comment 1. [..]  while also  reducing the costs…costs of?  

Response1. Thank you for pointing this out, edition has been done on this sentence. The Adolescent and Youth-friendly services (AYFS) Programme has the potential to address several diverse problems within adolescents' healthcare systems by improving the quality, accessibility, efficiency, and effectiveness of healthcare services.

Comment 2. […]  their success rate has been inconsistent…This is not true, AYFS has

been successful in other regions in SA, assisting to educate youth.

Response 2. Despite the potential for AYFS to improve the quality and accessibility of adolescent healthcare services, AYFS provision and impact has been evaluated and found to be relatively low in Soweto.  (Geary, R.S.; Webb, E.L.; Clarke, L.; Norris, S.A. Evaluating youth-friendly health services: young people's perspectives from a simulated client study in urban South Africa. Global Health Action 2015, 8, 26080).

Comment 3.  The objective of this study is to evaluate the effectiveness of the Adolescent and Youth-friendly services Programme against the WHO global standards for quality health-care services for adolescents in Vhembe district, Limpopo.

Response 3. Therefore, the objective of this study was to evaluate the implementation of AYFS against the World Health Organization (WHO) global standards for quality health-care services for adolescents to strengthen these services in Vhembe district, Limpopo.

Comment 4.  08 critical standards – write in words “eight”

Response 4. Thank you for pointing this out, edition has been done on this sentence. AYFS has been evaluated in depth across eight WHO global standards for quality health-care services for adolescents.

Comment 5.  critical standards….what are these critical standards? Is it the same as the mentioned WHO global standards? 

Response 5. Yes, however the word critical has been removed and add WHO global standards to  make it clear. AYFS has been evaluated in depth across eight WHO global standards for quality health-care services for adolescents.

Comment 6. I previously highlighted the benefit of using headings in your abstract so that you can see how to present the abstract succinctly. The current version needs to be strengthened.  

Response 6.  Agree. We have, accordingly, revised and strengthened the abstract with the use of sub-headings.

Comment 7. The methods in the abstract must mention what you did…cross-sectional study using questionnaires, what are the methods? how was data analysed

etc.  

Response 7. The methods in the abstract has been revised accordingly. A cross-sectional study was conducted to evaluate the implementation of AYFS against the WHO global standards for quality healthcare services for adolescents to strengthen these services in Vhembe district, Limpopo. Evaluating the implementation of AYFS was conducted through questionnaires distributed to healthcare providers in the selected primary healthcare facilities in Vhembe district. For descriptive statistical analysis, research data was analysed using Statistical Package for the Social Sciences (SPSS).

Comment 8. A rise in AYFS does not mean that AYFS has not been implemented effectively.

Response 8 Despite the potential for AYFS to improve the quality and accessibility of adolescent healthcare services, AYFS provision and impact has been found to be relatively low in Soweto [18].

Comments 9. The background needs to be strengthened. When talking about the effectiveness of AYFS, it looks like the focus is on HIV and if there’s been an increase or rise in HIV rate. However, AYFS seeks to address more than HIV. In the title the authors mention STI’S (which a multiple) and then teen pregnancy. But the background both in the abstract ad the introduction does not succinctly position  AYFS amidst all the sexual reproductive health issues that AYFS seeks contribute towards.  

Response 9. More than 1 million people are infected with sexually transmitted infections (STIs) each day throughout the world [4]. As a result, there are about 376 million cases of STIs every year, which include chlamydia, syphilis, trichomoniasis, and gonorrhea, among others [5]. In addition, it is estimated that more than 500 million people are living with genital infections around the world, with herpes simplex virus (HSV) being the most common [5]. The prevalence of these STIs among young women in Vhembe district is disproportionally higher than that among young men in the same age group [6]. It is believed that this increased risk is the result of a combination of factors, including gender inequity in (sexual) relationships, early sexual debut ages, and lack of access to sexual and reproductive health (SRH)[6].

Comments 10. There is still no clear objective of this study. It seems like an evaluation of how AYFs has been implemented BUT this is still not clearly written. Are the authors evaluating AYFS against the defined standards to inform initiatives for strengthening these services?  Yes

Evaluation is broad. One can have specific objectives in a overall evaluation. Example : (1) Assess the extent to which AYFS standards have been implemented in clinics that have adopted/implemented the AYFS programme. (2) Examine factors that influence uptake of AYFS at the district and zonal levels among adolescent girls (or use focus on the nurses that run AYFS in clinics. (3) Determine the

coverage of AYFS etc. 1

These are examples of how you become specific. So are you evaluating how AYFS in Vhembe clinics has contributed to educating adolescent girls (or boys) about HIV, STI’S and Teen pregnancy? Or are you evaluating the role of health care workers in following the eight WHO standards within the AYFS programme in the clinics.  

You need to narrow down and specify what THIS paper is aimed to do!  

Again, once this has been established, example, if your focus is the role of healthcare workers in following the standards. Then the introduction/literature review needs to align with that. The introduction does now provide a strong background for this study. No mention of healthcare workers and their role in sexual reproductive services etc.

Random sentences to edit…

Response 10. Yes, authors are evaluating AYFS against the defined WHO global standards to inform initiatives for strengthening these services.

Comments 11. Although, the Department of health have developed National 59 Adolescent & Youth Health Policy (AYHP) 2017, and National Adolescent Sexual and Reproductive Health and Rights Framework Strategy (ASRH&R) 2015 to improve the reproductive health of adolescents, AYFS have not been sufficiently funded or expanded in South Africa….No reference.  

Response 11. Noted, references are now provided accordingly. To curb new HIV infections and pregnancies among adolescents and teenage pregnancy, the National Department of Health (NDoH) introduced measures such as National Adolescent & Youth Health Policy (AYHP) [13], and National Adolescent Sexual and Reproductive Health and Rights Framework Strategy (ASRH&R) [14].

Comments 12. The majority of interventions have not addressed the specific needs of the young and adolescent population group due to low implementation and non standardized AYFS..what does low implementation mean?

Response 12. The majority of interventions have not addressed the specific needs of the young and adolescent population group due limited resources for AYFS implementation [22,23].

Comments 13. As part of the WHO recommendation that AYFS should be accessible, acceptable, equitable, appropriate, and effective, as well as providing quality assurance in implementing AYFS this sounds incomplete …edit

Response 13. As part of the WHO recommendation that AYFS should be accessible, acceptable, equitable, appropriate, and effective, as well as providing quality assurance in implementing AYFS to improve adolescents' health outcomes.

Comments 14. It is nevertheless imperative that, as with all other programs, the facilities implementing these services must be evaluated to identify and address any shortcomings and build upon the program's strengths and successes. Is that what you want to

do? Evaluate the shortcomings of AYFS in Vhembe?  

Response 14.  Vhembe district has not been evaluated regarding implementation of AYFS based on WHO global standards.

Comments 15. Vhembe district has not been evaluated regarding the implementation of AYFS based on WHO global standards. This seems to be focus, let this come from the beginning as the objective.  

Response 15. Noted, and the suggestions has been amended as the reviewers suggested.